# Awareness and Attitudes towards Advance Care Directives (ACDs): An Online Survey of Portuguese Adults

**DOI:** 10.3390/healthcare9060648

**Published:** 2021-05-29

**Authors:** Carlos Laranjeira, Maria dos Anjos Dixe, Luís Gueifão, Lina Caetano, Rui Passadouro, Ana Querido

**Affiliations:** 1School of Health Sciences of Polytechnic of Leiria, Campus 2, Morro do Lena, Alto do Vieiro, Apartado 4137, 2411-901 Leiria, Portugal; maria.dixe@ipleiria.pt; 2Centre for Innovative Care and Health Technology (ciTechCare), Rua de Santo André—66–68, Campus 5, Polytechnic of Leiria, 2410-541 Leiria, Portugal; rmfonseca@arscentro.min-saude.pt; 3Research in Education and Community Intervention (RECI I & D), Piaget Institute, 3515-776 Viseu, Portugal; 4Intensive Care Unit, Leiria Hospital Center, R. de Santo André, 2410-197 Leiria, Portugal; luisgueifao@gmail.com (L.G.); linamariacaetano@gmail.com (L.C.); 5Center for Research in Health and Information Systems (CINTESIS), NursID, University of Porto, 4200-450 Porto, Portugal

**Keywords:** advance care directives, attitudes, knowledge, cross-sectional study, Portugal

## Abstract

(1) Background: Evidence shows that facilitated advance decisions can increase the number of meaningful and valid Advance Care Directives (ACDs) and improve the quality of care when End-Of-Life (EOL) is near. Little is known about the awareness and attitudes of Portuguese adults towards ACDs. The present study aims to assess the knowledge, attitudes, and preferences of a sample of Portuguese adults regarding EOL care decisions and ACDs. (2) Methods: A total of 1024 adults were assessed with an online cross-sectional survey collecting information on sociodemographic factors, knowledge, attitudes and preferences regarding advance decisions and EOL care. (3) Results: Participants had a mean age of 40.28 ± 11.41 years. Most were female and had a professional background related to healthcare. While 76.37% of participants had heard of ACDs, only a small percentage (2.34%) had actually ever made an ACD. Knowledge levels were weakly correlated with attitudes regarding ACDs (r = −0.344; *p* < 0.01). (4) Conclusions: Participants lacked a comprehensive understanding about ACDs, but revealed positive attitudes towards their use and usefulness. Further research can inform efforts to improve ACD engagement in this population. The discussion about ACDs should be part of health promotion education with a focus on planning for a comfortable and peaceful death.

## 1. Introduction

Technological progress and advances in medical knowledge have made it possible to offer a longer life, although not always with the desired quality of life, such as the case of disproportionate use of treatment in terminally ill patients, therapeutic obstinacy, or dysthanasia [1]. In many situations of irreversible disease, modern medicine may prolong life at any cost, with great suffering and without benefit to the user, violating the right to a dignified life and death.

Thus, the concept of a patient’s right to self-determination is currently gaining importance and centrality when deciding what to do, while the traditional paternalistic posture of health professionals is hardly acceptable in the current context [1]. In terminally ill patients or in victims of accidents or sudden illness with irreversible injuries, determining the limits of medical intervention is increasingly the paradigm of action. Therefore, many questions arise regarding End-Of-Life (EOL) decisions, especially when patients are no longer able to express their will [1,2].

Advance Care Directives (ACDs) emerged from the need to reinforce the principle of self-determination in the context of significant medical-scientific developments, mainly to support human life when the patient is not in possession of their mental abilities. The ACD is the first legal tool to formally communicate one’s will regarding health care at the EOL [2]. An ACD indicates, usually as a written statement, a person’s preferences about medical treatments and decisions to be made when they are no longer mentally capable of communicating them, particularly regarding life-sustaining treatments that should be withheld/withdrawn [3].

According to Kermel-Schiffman and Werner [4], an ACD was associated with several benefits for the person, family, and professionals. The person exercises autonomy, affirms respect for their values and treatment preferences, improving their quality of life and life satisfaction at EOL, and facilitating patient-centred EOL care. Family members decrease their decision-making burden and worry about the future of the person and caregiver, easing stress, anxiety, and post-mortem depression. Lastly, health care professionals experience decreased moral distress [5].

Advanced care planning is an important and emergent topic in today’s public health agenda. ACDs’ effectiveness is internationally recognized, and their implementation, benefits, and obstacles have been extensively studied among different cultures and medical scenarios [6]. Research suggests that multiple factors affect EOL decision-making, including interactions among patients, family members, and health care providers [7]. Cultural factors can play an important role in enabling or hindering an ACD, as discussed in a systematic review by McDermott & Selman [7]. The authors found culture-related barriers to making an ACD, including mistrust between patients and clinicians, and cultural variation in willingness to discuss death. Portuguese culture is deeply rooted in the Catholic religion; for that reason, is upsetting to talk about death and dying [2]. A fatalistic point of view, characterized by a persistent melancholic and pessimist essence, is an essential trait of the Portuguese collective psyche. Fate and denial of death is a key part of the general lack of awareness of ACDs [2]. Attention to these cultural factors is particularly relevant because the concept of an advance care plan is not universally accepted and may be seen as intrusive or unnecessary [8].

Globally, the willingness to make an ACD is low. Studies indicate values between 18% and 34% in the US [9,10] and between 1.8% and 19% in Europe [11]. ACD prevalence in Australia, according to one study, was only 6% [12]. In Portugal, the first national study of community prevalence found completion rates of approximately 1.4% [13].

Spain, Finland, Holland, Belgium and Germany were some of the European countries that first enacted regulatory legislation on ACDs [14]. In Portugal, the debate around ACD legislation only began in 2006, following a legislative proposal prepared by the Portuguese Bioethics Association. However, the law was only approved by Parliament on 1 June 2012, and published on July 16 of the same year [14]. The Portuguese Living Will Law has a triple objective: (i) to establish the ACD’s legal regime; (ii) to regulate the appointment of the healthcare proxy; and (iii) to create the Living Will National Registry (RENTEV), the formal ACD registry [15]. This digital platform is an information system developed by the Portuguese Health Ministry to receive, register, organize and update all information and documentation regarding ACDs and health care proxies for all national, foreign, and stateless citizens residing in Portugal [16].

During the past 20 years, issues related to EOL care have attracted international attention, including high-profile legal cases and debates about advance care planning [17,18,19]. Much of the research has focused on the awareness, attitudes, and experience of health and social workers regarding ACD [3,20,21]. The evidence shows a lack of knowledge and misconceptions about ACD among these professionals [3,21]. Few studies have been conducted on public perspectives regarding EOL issues, including attitudes towards ACD. Better public awareness and openness to discuss death and EOL issues is needed to improve ACD completion [5]. However, there is a lack of community-based data regarding the factors associated with ACD completion among adults aged 18 or older. The present study was motivated by a widespread feeling, formed through opinions and news about RENTEV data, that adherence to ACD is low in Portugal, although several studies indicate that most people would like their wishes respected. In addition, the present research took place during an intense debate in Portuguese society regarding EOL options, in particular around euthanasia and assisted suicide [22,23].

Applying a socioecological approach, this study aimed (a) to ascertain the knowledge, attitudes, and preferences of a sample of Portuguese adults regarding EOL care and ACD; (b) to identify the sources of information used to obtain knowledge; and (c) to understand the relationship between knowledge and attitudes towards ACD.

## 2. Materials and Methods

### 2.1. Study Design and Participants

An online cross-sectional survey using a convenience sampling method was conducted between March and May 2018. Participants accessed the link to the online questionnaire through various social media platforms and emails. To ensure transparency in the study design and recruitment process, only adult men and women (aged ≥ 18 years) of Portuguese nationality were recruited, through an invitation via Messenger and Facebook to participate in an online survey. The survey was designed using www.googleforms.com. To avoid duplications or fraud, respondents were required to pass a CAPTCHA test, use cookies to detect multiple submissions, and delete the survey’s link from their Facebook page and Messenger account after conclusion [24]. Within two weeks of survey activation, we received 254 responses, of which 54 were deemed incomplete and excluded. The invitation to participate was also emailed to 1400 potential participants using the IPLeiria mailing list. Participants were asked to return the questionnaire within two weeks, and we received 824 valid responses. Thus, overall, 1024 participants completed the survey with a response rate of 64%.

This study was reported following the Strengthening the Reporting of Observational Studies in Epidemiology (STROBE) checklist.

### 2.2. Instruments

The survey instrument included three sections. Due to the topic’s sensitivity, sections two and three were optional, and participants were given the choice of answering the full questionnaire.

Section one included questions related to personal information, demographic data (age, gender, marital status, highest level of education, occupation, religious beliefs), and information about ACDs. As the experience of extreme situations, such as death or great suffering, alters our perception of the EOL, we sought to know who had already experienced the death of family members or loved ones, as well as who had already experienced situations of palliative care with family or friends. Three questions were added to explore if individuals had heard of ACD, what sources they used to obtain information about ACD, and whether they had made an ACD.

Section two consisted of the General Public Attitudes Toward Advance Care Directives (GPATACD) Scale. This instrument was used to examine the attitudes about advance directives. Consisting of 26 items, the GPATACD is a 5-point Likert scale (‘strongly disagree’ = 1 and ‘strongly agree’ = 5). Items 1, 2, 4, 5, 6, 9, 10, 12, 15, 16, 19, and 22 were inverted in the decreasing direction (lower values, more positive attitudes). The scale had a four-factor structure: ‘autonomy and dignity of the person at the EOL’, ‘EOL decision-making’, ‘application of ACD’, and ‘perception of the EOL’. This scale was originally developed by Laranjeira et al. [25] and has a good reliability and validity. The Cronbach α of the GPATACD measure used in this current study was 0.848.

Section three consisted of questions pertaining to a participant’s knowledge about advance directives. This Advance Directives Knowledge Scale (ADKS) included 16 items (true/false), where incorrect and correct answers are given a score of zero and one, respectively. Thus, totals range from 0 to 16, with a higher score indicating better knowledge. Statements were constructed based on previous literature [26,27]. The internal consistency of the instrument was analysed using the Kuder–Richardson Formula 20 (KR-20) coefficient. According to Hinton et al. [28], a KR-20 coefficient between 0.50 and 0.70 indicates moderate reliability; thus, the current study’s KR-20 of 0.51 reached an acceptable level.

Before conducting the study with the full sample, the survey was applied to a small number of subjects (n = 15) in order to determine the appropriateness of language, survey length, and ease of understanding of each item. The instrument took 5–10 min to be completed, and participants showed satisfaction with the questionnaire length. Each item was understood, rated as relevant, appropriate, not difficult, and correctly interpreted by >80% of participants. Given the results of the pilot study, the overall format of the questionnaire remained unchanged (some survey items suffered minor wording modifications).

### 2.3. Formal and Ethical Procedures

Written approval to conduct the study was obtained from the dean, research board, and the host institution’s ethics review board (IPLeiria, approval n. 3/2018). All procedures in this study were performed in accordance with the Helsinki Declaration and its later amendments or comparable ethical standards. Firstly, an anonymous online survey was developed to ensure that personal information was kept confidential. Secondly, the participants were clearly informed that participation was voluntary, their responses would be confidential and kept secure, and data would only be used for academic purposes. Before data collection, a signed informed consent was secured, assuring participant anonymity and confidentiality. No monetary rewards were given for completing the questionnaire.

### 2.4. Data Analysis

The data were analysed using SPSS version 26 (IBM, Armonk, NY, USA). The mean and Standard Deviation (SD) were computed for the numeric variables, whereas the frequency and percentage were computed for the categorical/nominal variables. Inferential statistical analysis was performed using Pearson’s correlation coefficient (quantitative variables). A *p* ≤ 0.05 was taken as statistically significant.

## 3. Results

### 3.1. Demographics

In total, 1024 participants participated in the online survey. Ages ranged from 18 to 78 years (mean = 40.88; SD = 11.41), corresponding to an adult population of working age. The majority of the participants were female (79.69%) and were married or living together (61.62%). Most participants (79.89%; n = 818) had a higher education. Considering the importance of religion in these types of questions, we found a high percentage of Catholics, 82.62% (n = 846). The majority (71.09%) had a healthcare professional background (Table 1).

### 3.2. Information about ACDs

Few participants had not experienced death situations (10.45%; n = 107), and nearly a third (30.66%; n = 314) had already had contact with palliative care situations. As the ACD is a little-publicized topic, we surveyed whether individuals had already heard about ACDs, where they had obtained information, and whether they had made an ACD. Only 23.63% (n = 242) of participants had never heard about the subject. Information was usually obtained from the Internet, known health care professionals, and newspapers and magazines. The family doctor was the source of information on ACDs for 20 individuals (1.63%). We emphasize that although the majority (76.37%) had heard of ACDs, only a small percentage (2.34%) had ever completed one.

### 3.3. Knowledge and Attitudes about the EOL Care and ACDs

As aforementioned, before advancing to sections two and three assessing knowledge and attitudes about EOL care, individuals were asked about their willingness and availability to answer questions about death, thus anticipating constraints or less comfortable situations. Participants were given the possibility of finishing and validating section one of the questionnaire, but answer no more questions. Only 9.38% (n = 96) of participants chose not to continue; thus, the final sample consisted of 928 interviewed individuals. In order to gauge knowledge about ACDs, participants were asked to classify 16 statements as true or false. A total of eight was considered the neutral value, above which knowledge was positive, and below which was negative. The minimum score was four right answers (0.22%; n = 2), and 6.47% (n = 60) obtained the mid-score of eight (Figure 1). Although around 71% of participants had a health care professional background, only 2 participants (0.2%) obtained the maximum of 16 correct answers, and 93.86% (n = 871) obtained a score below 13 correct answers.

Several statements had a high percentage of correct answers (Table 2). Nearly all participants (98.71%) correctly indicated a distinction between ACDs and euthanasia. An expressive majority demonstrated adequate knowledge about the primacy of ACDs over family opinion (96.77%) and whether ACD enforcement requires family approval (95.47%), indicating an excellent level of knowledge regarding the role of the family in relation to ACDs. There was a high percentage of incorrect answers to the question related to organ donation and ACDs (61.31%; n = 569), revealing a lack of discrimination between both topics. Regarding the formal process, 90.30% (n = 838) considered that ACDs are only valid if registered in the RENTEV, evidencing ignorance by a great majority of the sample concerning ACD validity. Regarding confidence in health care, 27.05% (n = 251) incorrectly indicated that ACDs protect doctors from being accused of negligence. There was some ignorance regarding the Health Care Prosecutor (HCP), with 42.46% (n = 394) considering that their appointment is mandatory whenever an ACD is made, and 94.61% (n = 878) indicating they did not know the function of the HCP.

The overall average obtained with the GPATACD scale, where lower scores correspond to more favourable attitudes, was 1.92, suggesting a positive attitude towards ACDs (Table 3). The GPATACD item with the lowest score, revealing a very favourable attitude, was item 2 (‘My opinion should not be respected in the EOL process’), with an average of 1.22 ± 0.51 points. The items with the highest scores, indicating less favourable attitudes towards ACDs, were item 27 (‘I have no information on ACD/Vital Testament’), with 2.83 ± 1.21 points; item 30 (‘I don’t want to think that I will eventually die or become disabled, to the point of not being able to make decisions’), with 2.64 ± 1.22 points; and item 29 (‘I do not make a Vital Testament because there is still little information available’), with 2.60 ± 1.10 points.

In addition, we examined the relationship between attitudes about ACD and the level of knowledge demonstrated (Table 4). There was a weak, negative, and significant correlation (*p* < 0.01) for three factors, while there was no correlation with the factor “Application of ACD” (*p* > 0.01). Overall, there was a negative, statistically significant correlation between the attitudes of participants towards ACDs and their level of knowledge; that is, higher levels of knowledge were associated with more favourable attitudes regarding ACDs.

## 4. Discussion

Portuguese legislation on ACD emerged in 2012, recognizing the right of each person to express their will—in advance and in a conscious, free and informed way—on whether to receive health care when they are unable to decide (Basic Law of Palliative Care—Law No. 52/2012 of September 5). As this is a recent topic, the study sample was asked if they had heard about ACDs and if they had made an ACD. The results showed that 23.63% were unaware of what an ACD was. However, these results are very positive when compared to other studies. In fact, in the study by Chung et al. [29], 85.7% of the sample had not heard about ACD, and similar results were found in the study by Ko et al. [30] and Gao et al. [31], respectively, 65% and 79%.

Among the 76.37% who had already heard about ACDs, 26.79% revealed that they had obtained information from sources not covered in the questionnaire answer options, although it was unclear what other sources were used. The next most-cited information sources on ACD were the Internet and health professionals. These results are in line with results on health information sources, most likely for the reasons previously described. In the study by Andrés-Pretel et al. [32], participants stated that the greatest source of information about ACD was the media.

Only 2.34% of the sample had an ACD. This number is higher than the general population, which according to the Portuguese National Health Service, was around 0.28% in January 2020 [33]. However, this is lower than that reported in the study by Gao et al. [31], which despite its low percentage of participants with ACD (only 10%) had a higher value than the present study.

The vast majority of the sample (89.55%) had already experienced situations of death of family members or loved ones, but 10.45% did not have this experience. It is also noteworthy that 69.34% of the participants had not experienced situations of family members/friends in palliative care. These results should be considered when analysing the remaining results because, as White et al. [12] revealed, the experience of situations close to death or suffering changes perceptions of the EOL. In addition, Andrés-Pretel et al. [32] recognized a better attitude towards ACD among people who had accompanied family members or people at the end of their lives.

From an initial sample of 1024, 9.38% decided to end the questionnaire after the first section, leaving this study with a final sample of the second and third parts composed of 928 participants. Although death may be a subject that is avoided, ignored, or denied, only a small percentage of the initial sample refused to address the issue. This fact provides further evidence against death as a ‘taboo’ conversation topic [34,35]. In a Canadian population survey [36], only 9% agreed that ‘end of life is too sensitive a topic to talk about’. The high percentage of health professionals in our sample may explain a greater predisposition to address a theme that is part of their daily work. On the other hand, all the participants who were unable to address the subject had various trainings and professions, but all of them were unrelated to healthcare.

Our results revealed good values of knowledge about ACDs. These results, conditioned by the sample’s socio-professional characteristics, differ from results in other studies carried out on the subject. Rossini et al. [37] and Silva et al. [38] concluded that there was a low level of knowledge about ACDs among Brazilian health professionals. Chung [29] and Ko [30] studies also revealed low levels of knowledge about ACDs.

The good level of knowledge and attitudes in this study may be explained by the fact that the vast majority of the sample were health-related professionals. This fact is supported by Barbosa’s study [39] that found adequate levels of knowledge among Portuguese medical students. Overall, the studies indicate that higher knowledge among healthcare professionals were associated with higher education and more years of working experience [4]. Nonetheless, another study in Portugal reaffirmed that health professionals displayed poor knowledge of ACDs, but a favourable attitude toward their usefulness [40]. Similar results were found in Spain [41,42]. People turn most often to health professionals for information, so they should be required to have a good or excellent knowledge level.

The meaning of death and dying and, consequently, the attitudes and practices at EOL and use of ACDs are also strongly influenced by cultural dogmas, sociocultural beliefs, and values [32]. Within contemporary western medicine, there is a growing recognition of the significant links between spirituality/religion and health and the need for health professionals to understand their clients’ spiritual/religious beliefs and practices. For instance, respect for the sanctity of life is a crucial value in Christian societies, which may influence public knowledge and attitudes toward ACD. Influenced by Catholicism, Portuguese perceptions about death and dying may be a significant obstacle to implementing and using ACDs [2].

In our study, higher levels of knowledge were associated with more positive attitudes towards ACD. These results are corroborated by Chung et al. [29], wherein greater prior knowledge about ACDs increased the willingness to carry out an ACD. Filing an ACD is not mandatory, and whether everyone should make one is still open for debate, but clearly everyone should learn about them to make an informed decision [43].

### 4.1. Methodological Issues

The study included a large sample size and used a culturally sensitive questionnaire, with acceptable levels of reliability and validity. However, our findings were limited by the use of convenience sampling and the inequalities in general characteristics of participants, which mainly included women with a professional background related to healthcare. As a result, some of our findings may be biased because this may have affected the relationship between attitudes and other factors towards ACDs. In turn, we cannot exclude the possibility of self-reported bias and socially desirable responses in the survey.

The data collected from survey respondents was not adjusted to represent the population from which the sample was drawn. Nonetheless, they can provide valuable clues for further studies based on random samples. Upcoming studies could estimate the knowledge, attitudes, and preferences of general adult population on a larger scale to be able to design proper interventions on a nationwide level.

People who recognize the need for an ACD are generally aware of its benefits, believe that ACDs can positively affect care at the EOL, and may be more apt to complete ACDs. Long-term studies are needed to determine the extent to which ACDs affect EOL care, and to identify factors that hinder ACDs from positively affecting care. Lastly, the part of the questionnaire asking about EOL care and ACD was optional, so only participants who agreed to respond to that part were included in the analysis. These subjects might have had more positive attitudes towards EOL issues, thereby introducing a bias in the results. Additionally, the strategies used to preserve data integrity following recruitment through social media were limited. Because our research adopted a cross-sectional design, the results cannot be generalized. Future studies would benefit from an in-depth qualitative approach exploring experiences regarding ACDs.

### 4.2. Implications

Educational materials or media campaigns might be helpful to encourage EOL discussions. Thus, health care professionals should attempt to increase public knowledge of ACDs by providing a comprehensive explanation of EOL decisions and ACDs. However, some studies using written materials without direct counselling were relatively ineffective in increasing ACD completion [44,45]. Indeed, providing oral information over multiple sessions was found to be the most successful intervention [46]. Thus, face-to-face discussions should be encouraged. To support such discussions, health professionals might need training. Besides specific steps to facilitate more meaningful conversations regarding EOL issues, important measures include public support for a range of educational initiatives and providing palliative care training to all healthcare professionals involved in delivering EOL care [3,47]. Health care professionals will need to address health literacy issues to evaluate educational interventions, their limits, enablers, and efficacy. Evaluations can therefore assess how interventions can mitigate barriers and leverage targeted enablers for a more significant effect [21,45]. Creating a person-centred foundation in all health care levels and processes is another key component, which—in practice—is a demanding aspiration.

The COVID-19 pandemic has also highlighted the importance of advanced care planning. It presents opportunities to widen understanding of its benefits, as there is raised public awareness of how health can deteriorate suddenly and unexpectedly [48].

## 5. Conclusions

To our knowledge, this online survey is one of the first large scale studies in Portugal regarding knowledge, attitudes, and preferences of advanced decisions and EOL care. ACDs were created to promote autonomous EOL decision-making, but they are not well understood or used by the Portuguese adults. The discussion about ACDs should be part of the health promotion taught by health professionals and should be presented to the general public as a normal part of health care, with a focus on planning for a comfortable and peaceful death. Community-based initiatives are a promising roadmap for bringing EOL planning options to a broad audience, tailoring programs to meet the distinctive social, cultural, and religious needs of the particular populations they serve. Health care professionals must find culturally sensitive methods for educating the population about ACD.

## Figures and Tables

**Figure 1 healthcare-09-00648-f001:**
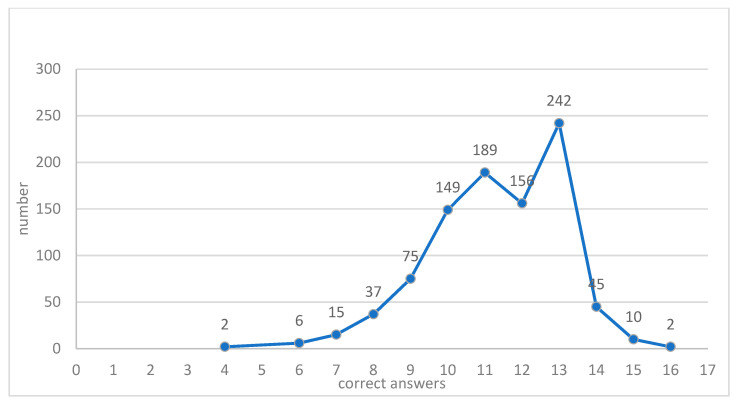
Frequencies of correct answers in the Advance Directives Knowledge Scale (ADKS).

**Table 1 healthcare-09-00648-t001:** Sample characteristics.

Characteristics	n (%)
**Gender**	
Male	208 (20.31)
Female	816 (79.69)
Age (mean/SD)	M = 40.28; SD = 11.41
**Level of Education**	
Elementary education	36 (3.52)
Secondary education	170 (16.60)
Higher education	818 (79.89)
**Marital status**	
Married/living together	631 (61.62)
Single	294 (28.71)
Divorced/separated/widowed	99 (9.67)
**Professional status**	
Healthcare professionals	728 (71.09)
Non-healthcare professionals	296 (28.91)
**Religious beliefs**	
Catholic	846 (82.62)
Protestant	9 (0.88)
Jehovah’s Witnesses	8 (0.78)
Jewish	1 (0.1)
No religion	148 (14.45)
Other religion	12 (1.17)

**Table 2 healthcare-09-00648-t002:** Sample distribution for each ADKS item.

		n	%
The ACD, once signed, is valid for life. *	Correct	695	74.89
	Incorrect	233	25.11
ACD and euthanasia mean the same thing. *	Correct	916	98.71
	Incorrect	12	1.29
The ACD, once made and registered, can be revoked at any time.	Correct	863	93.00
	Incorrect	65	7.00
The ACD is valid as an authorization or refusal to participate in scientific research programs or clinical trials.	Correct	565	60.88
	Incorrect	363	39.12
The ACD gives an indication of whether the citizen allows their organs to be donated or not.	Correct	359	38.69
	Incorrect	569	61.31
The ACD reflects the values and preferences of citizens when making therapeutic decisions at the end of their lives.	Correct	861	92.78
	Incorrect	67	7.22
The ACD defends doctors from accusations of negligence.	Correct	677	72.95
	Incorrect	251	27.05
The ACD indicates the citizen’s clear and unequivocal will to not be subjected to artificial support treatments for vital functions.	Correct	872	93.97
	Incorrect	56	6.03
The ACD guarantees the citizen’s choice to not be subjected to experimental treatments that are in an experimental phase.	Correct	718	77.37
	Incorrect	210	22.63
Appointing a health care prosecutor is mandatory whenever a living will is made. *	Correct	534	57.54
	Incorrect	394	42.46
The appointment of the health care prosecutor replaces the ACD. *	Correct	50	5.39
	Incorrect	878	94.61
The ACD is only effective if registered in the Living Will National Register (RENTEV).	Correct	90	9.70
	Incorrect	838	90.30
The patient’s family has to agree with the content of the ACD statement so it can be applied. *	Correct	886	95.47
	Incorrect	42	4.53
To be fulfilled, the ACD must always accompany the person or be taken by someone to the hospital.	Correct	669	72.09
	Incorrect	259	27.91
The family’s opinion overrides that of the health care prosecutor. *	Correct	866	93.32
	Incorrect	62	6.68
When the patient is unconscious, their family can change or cancel the content of the ACD. *	Correct	898	96.77
	Incorrect	30	3.23
Total of each statement		928	100

* Wrong statements.

**Table 3 healthcare-09-00648-t003:** Descriptive statistics of the General Public Attitudes toward Advance Care Directives (GPATACD) Scale.

	M ** (SD ^†^)
1. The existence of the vital testament is not important. *	1.56 (0.74)
2. My opinion should not be respected in the EOL process. *	1.22 (0.51)
4. The ACD does not reflect the patient’s values and preferences when making therapeutic decisions at the EOL. *	1.88 (0.84)
5. ACDs are a useful tool for healthcare professionals when making decisions about EOL patients. *	1.76 (0.85)
6. The health care prosecutor appointed by the patient does not facilitate the professionals’ decision-making. *	2.18 (0.83)
7. Compliance with ACDs pertain to the physician.	1.79 (1.03)
8. ACDs are a legal form of euthanasia.	1.56 (0.96)
9. It is not important that patients make their vital testament or ACD. *	1.73 (0.87)
10. It is not important that all citizens make their vital testament or ACD. *	1.79 (0.92)
11. ACDs are important only for religious reasons.	1.33 (0.73)
12. Legalization of the vital testament did not contribute to human dignity. *	1.61 (0.83)
13. Death must be postponed, regardless of the person’s condition.	1.58 (0.95)
14. EOL care should be provided based on the opinion of the health professional.	2.54 (1.13)
15. EOL care should not be provided based on the patient’s opinion. *	2.16 (0.98)
16. I do not want to be able to have an opinion on the care I can receive in an end-of-life situation. *	1.40 (0.64)
17. EOL care should be provided based on the opinion of the family.	2.11 (1.01)
18. My family will make the EOL decisions for me, when necessary.	2.19 (1.14)
19. I will overwhelm my family with EOL decisions. *	1.66 (0.93)
20. My doctor will make the EOL decisions for me when the time comes.	2.07 (1.11)
22. The vital testament is only important for elderly and sick people. *	1.51 (0.78)
25. I am currently healthy, but there may be a need to consider decisions regarding the final phase of my life.	2.08 (1.17)
26. At my current age, there is no need to consider EOL decisions.	1.85 (1.04)
27. I have no information on ACD/vital testament.	2.83 (1.21)
28. It is possible to make EOL decisions, even if I cannot imagine myself in such a situation.	2.26 (1.05)
29. I do not make a Vital Testament because there is still little information available.	2.60 (1.05)
30. I don’t want to think that I will eventually die or become disabled, to the point of not being able to make decisions.	2.64 (1.22)
Total	1.92 (0.44)

* Negative items; ** M = Mean; ^†^ SD = Standard Deviation.

**Table 4 healthcare-09-00648-t004:** Pearson’s correlation between the attitudes towards ACD (GPATACD) and the level of knowledge (ADKS).

	Knowledge (ADKS)
Autonomy and dignity of the person at the EOL	−0.257 ^§^
EOL decision-making	−0.384 ^§^
Application of ACD	0.007
Perception of the EOL	−0.246 ^§^
Total—Attitudes toward ACD	−0.344 ^§^

^§^ The correlation is significant at the 0.01 (bilateral) level.

## Data Availability

All data generated or analyzed during this study are included in this article.

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
