# Peer review of "Awareness and Attitudes towards Advance Care Directives (ACDs): An Online Survey of Portuguese Adults"

_healthcare, 2021, doi:10.3390/healthcare9060648_

Round 1

Reviewer 1 Report

The authors have faithfully revised the paper according to the content of the review. I would like to agree to the publishing as it is.

Author Response

We thank the reviewer for the careful reading and positive comments.

Reviewer 2 Report

I have to congratulate the authors for their work, since it has been highly improved and highlighted a really interesting topic. 

Author Response

(The authors gave the same response as above.)

Reviewer 3 Report

This is an interesting study on an important and current topic. The authors did a good job, but I find a significant problem. As the authors themselves state in the "limitations" section, the use of convenience sampling does not allow conclusions to be drawn about end-of-life care and ACD in the entire Portuguese adult population. Statistical sampling would be necessary to achieve the objectives set by the authors.

Author Response

This is an interesting study on an important and current topic. The authors did a good job, but I find a significant problem. As the authors themselves state in the "limitations" section, the use of convenience sampling does not allow conclusions to be drawn about end-of-life care and ACD in the entire Portuguese adult population. Statistical sampling would be necessary to achieve the objectives set by the authors.

Response: We really appreciate the reviewer´s suggestion. Following the recommendation, the manuscript has been revised, improving the objectives and the methodological issues/ conclusion subsections. Please see the track changes activate.